# LRG1 Associates with Iron Deficiency Anemia Markers in Adolescents

**DOI:** 10.3390/nu15143100

**Published:** 2023-07-11

**Authors:** Rashed Alhammad, Mohamed Abu-Farha, Abdur Rahman, Thangavel Alphonse Thanaraj, Lemia Shaban, Reem Alsabah, Samar Hamad, Maha M. Hammad, Arshad Channanath, Fahd Al-Mulla, Jehad Abubaker

**Affiliations:** 1Department of Pharmacology, Faculty of Medicine, Kuwait University, Kuwait City 13110, Kuwait; rashed.alhammad@ku.edu.kw; 2Biochemistry and Molecular Biology Department, Dasman Diabetes Institute, Kuwait City 15462, Kuwait; mohamed.abufarha@dasmaninstitute.org (M.A.-F.); hamad.samar@gmail.com (S.H.); maha.hammad@dasmaninstitute.org (M.M.H.); 3Department of Food Science and Nutrition, College of Life Sciences, Kuwait University, Kuwait City 13110, Kuwait; abdurrahman.ahmad@ku.edu.kw (A.R.); dr.lemiashaban@gmail.com (L.S.); 4Department of Community Medicine and Behavioural Sciences, Faculty of Medicine, Kuwait University, Kuwait City 13110, Kuwait; alphonse.thangavel@dasmaninstitute.org (T.A.T.); reem1@hsc.edu.kw (R.A.); 5Genetics and Bioinformatics Department, Dasman Diabetes Institute, Kuwait City 15462, Kuwait; arshad.channanath@dasmaninstitute.org

**Keywords:** LRG, iron deficiency anemia, Hb, HIF-1α, ferritin

## Abstract

Leucine-rich α-2 glycoprotein1 (LRG1) has been shown to be associated with several health conditions; however, its association with iron deficiency anemia, especially in children, has not been previously explored. In this study, we investigated the association between LRG1 and several iron deficiency anemia markers, including hemoglobin (Hb), albumin, red cell distribution width (RDW), iron, ferritin, and Hb transferrin saturation. A total of 431 participants were included in this analysis aged between 11 and 14 years. Higher LRG1 levels were observed in children diagnosed with anemia [31.1 (24.6, 43.2) µg/mL] compared to non-anemic children [29.2 (22.7–35.95) µg/mL]. Statistically significant differences of LRG1 level across the three groups (tertiles) of Hb, iron, transferrin saturation, albumin, RDW, ferritin, and WBC were observed. Strong negative correlations were observed between LRG1 and Hb (Spearman’s rho = −0.11, *p* = 0.021), albumin (Spearman’s rho = −0.24, *p* < 0.001), iron (Spearman’s rho = −0.25, *p* < 0.001), and Hb transferrin saturation (Spearman’s rho = −0.24, *p* < 0.001), whereas circulating LRG1 levels were positively associated with RDW (Spearman’s rho = 0.21, *p* < 0.001). In conclusion, our findings demonstrate for the first time the strong association between iron deficiency anemia markers and LRG1 in otherwise healthy school-aged children. However, further studies are needed to corroborate those results and to look for similar associations in other population subgroups.

## 1. Introduction

Anemia is a major health problem, especially in developed countries, where it affects about 40% of children [1]. In addition to the individual health impact, anemia has a negative impact on countries’ development as it is associated with poor cognitive and motor development in children, as well as reduced work capacity in adults [2,3]. A combination of nutritional, genetics, or infection factors can lead to the development of anemia. Anemia is caused by decreased erythrocyte production or increased blood loss [4]. Iron is an essential metal that is involved in multiple cellular processes including oxygen transport, DNA synthesis, and enzymatic activities [5]. Key proteins affecting iron metabolism and transport are transferrin and ferritin [6]. Transferrin regulates the transport of iron in the bloodstream, delivering it through the body, while ferritin mediates the storage of excess iron within cells [7]. Iron deficiency orchestrates iron regulation through the iron regulatory proteins (IRPs) which bind to the 3′ UTR of transferrin receptor (TfR1) or 5′ UTR of ferritin mRNA, leading to increased iron uptake or decreased iron storage, respectively [8].

Leucine-rich α-2 glycoprotein 1 (LRG1) is a leucine-rich repeat protein that has been associated with many diseases including cancer, heart diseases, and diabetes [9]. It has been shown to play a role in metabolic pathways related to diet-induced hepatosteatosis, insulin resistance, and angiogenesis [10]. LRG1 has been shown to be a pro-angiogenic factor, enhancing angiogenesis through the activin receptor-like kinase 1 (ALK-1) signaling pathway [11]. LRG1 has been shown to increase angiogenesis in ocular diseases and in colorectal cancer, by promoting the expression of vascular endothelial growth factor (VEGF) in a Hypoxia-inducible factor-1 alpha (HIF-1α)-dependent fashion [12,13]. Moreover, LRG1 has been shown to regulate apoptosis and autophagy by activating the HIF-1α pathway and regulating HIF-1α protein expression [14]. Iron deficiency has also been shown to increase HIF-1α expression, with subsequent regulation of VEGF [15].

Overall, these previous reports indicate that both LRG1 and iron deficiency are implicated in regulating the HIF-1α pathway. However, the association between LRG1 and iron deficiency in children has not been well established. In this report, we explored the association between LRG1 and anemia in children by assessing the correlation between LRG1 and several iron deficiency markers, including hemoglobin, iron, percentage of transferrin saturation, ferritin, albumin, and red cell distribution width (RDW).

## 2. Material and Methods

### 2.1. Study Participants, Ethics, Consent, and Permissions

This study was conducted in 12 public middle schools selected from across all the six governorates of the State of Kuwait as previously described [16,17]. The study was approved by the Institutional Ethics committee at Dasman Diabetes Institute No. RA HM-2017-026 and the Ethics Committee at the Ministry of Health, Kuwait No: 2015/248). Written informed consent was obtained from the parents, as well as assent of each study participant. A total of 431 participants were included in this analysis, aged between 11 and 14 years. Students with chronic diseases and hemoglobinopathies were excluded from the study. We certify that the work conducted in this research complies with the ethical standards recommended by the Helsinki Declaration.

### 2.2. Blood Collection and Biochemical Analyses

A 5 mL blood sample was collected from the participants by venepuncture, and the plasma was separated and stored at −80 °C for later analysis. Full blood analysis was performed using a Beckman Counter Unicel DxH 800 hematology analyzer (Beckman Coulter Inc., Fullerton, CA, USA). Anemia was diagnosed according to the WHO criteria where males and females younger than 12 years with Hb less than 115 g/L were considered anemic, while 12–14 year old participants with Hb concentration < 120 g/L were considered anemic.

### 2.3. ELISA Assays for LRG1

Plasma samples were analyzed for LRG1 concentrations using an ELISA Kit (Cat# 27769. IBL, Gunma, Japan) with optimal dilution 1:4000 and following the manufacturer’s instructions.

### 2.4. Statistical Analysis

Data were analyzed for 431 subjects. Descriptive statistics of sample characteristics were performed and reported as median (interquartile range: IQR). Pearson’s chi-squared test, Fisher’s exact test, and the Wilcoxon rank sum test were run to determine significant differences between the groups. Spearman’s correlation was used to ascertain the association between LRG1 and Hb, iron, albumin, transferrin saturation, RDW, and ferritin. Subjects were also grouped according to the tertiles (two points that divide an ordered distribution into three parts) of markers of interest. The LRG1 levels across these tertiles are compared using the Kruskal–Wallis test. A *p* value < 0.05 was considered as statistically significant. All statistical analyses were performed using R statistical software (R Core Team, 2020).

## 3. Results

### 3.1. Description of the Study Group in School Adolescents

The demographic characteristics of the study group are summarized in Table 1. The mean (SD) age of the population was 12.3 (0.8) years. Most (234/431, 54%) of the participants were females while (197/431, 46%) were males. The median Hb and Hematocrit test (Hct) levels were 132 g/dL and 0.404 L/L, respectively. We also checked for gender-specific differences in the levels of RBC, WBC, Hb, Hct, and Gluc. Of these five markers, only Hb and RBC showed sex-specific differences at a *p*-value < 0.05 (Appendix A). We also assessed the distribution of anemia subjects based on gender (Appendix A) and the result was not significant.

### 3.2. Elevation of Circulatory LRG1 Level in Anemia

Our data show that plasma LRG1 levels were increased in children diagnosed with anemia [31.1 (24.6, 43.2) µg/mL] compared to non-anemic children [29.2 (22.7–35.95) µg/mL] (*p* = 0.07). Table 2 shows that there was a statistically significant difference in LRG1 level across the three groups (tertiles) of Hb (Figure 1A), iron (Figure 1B), transferrin saturation (Figure 1C), albumin (Figure 1D), RDW (Figure 1E), and ferritin (Figure 1F). The LRG1 level was significantly lower in the higher tertiles of albumin and transferrin saturation compared to the middle and lower tertiles. LRG1 level did not differ significantly between higher and middle tertiles of iron (*p* = 0.054); however, the difference was significant between higher and lower tertiles (*p* < 0.01). The LRG1 level was significantly higher in the higher tertiles of RDW, ferritin, and WBC compared to the middle tertiles.

### 3.3. LRG1 Associates with Iron Deficiency Anemia Markers

We found a strong negative correlation between LRG1 and Hb (Spearman’s rho = −0.11, *p* = 0.021; Figure 2A), and albumin (Spearman’s rho = −0.24, *p* < 0.001; Figure 2B) levels. On the other hand, circulating LRG1 levels were positively associated with RDW (Spearman’s rho = 0.21, *p* < 0.001; Figure 2C). Moreover, there was a negative correlation between LRG1 level and both iron (Spearman’s rho = −0.25, *p* < 0.001; Figure 2D) and Hb transferrin saturation (Spearman’s rho = −0.24, *p* < 0.001; Figure 2E). A statistical trend was also observed between the LRG1 level and ferritin (Spearman’s rho = −0.092, *p* = 0.064; Figure 2F). We also assessed the associations between MCV and LRG and the result was not statistically significant (Appendix A). Furthermore, evaluation of various clinical characteristics based on anemia status was conducted (Appendix A).

## 4. Discussion

This study aimed to examine the possible correlation between LRG1 and one of the most common diseases affecting children in developing countries, namely iron deficiency anemia. Iron deficiency can affect the general health as well as cognition and motor development [3,18,19,20]. It would therefore be of great advantage to identify novel screening markers for its early detection. While the exact relationship between LRG1 and iron deficiency anemia remains largely unclear, one hypothesis is based on its involvement in the regulation of HIF-1α protein expression, because it has been shown that iron deficiency increases HIF-1α which regulates VEGF. Moreover, LRG1 has been shown to activate the HIF-1α pathway and regulate HIF-1α protein expression [14]. Given that both LRG1 and iron deficiency regulate HIF-1α expression, it is vital to explore the association between LRG1 and anemia. In this report, we explored the association between LRG1 and anemia in school-aged children by assessing the correlation between LRG1 and several iron deficiency markers, including Hb, iron, percentage of transferrin saturation, albumin, and RDW.

Several published reports indicate that serum iron and transferrin saturation are biochemical markers of iron status that are used in iron deficiency diagnosis [21,22,23]. In addition, albumin has been shown to be negatively associated with iron deficiency [24] and is considered a predictor of iron deficiency [25]. Moreover, the transferrin/albumin ratio has also been used in iron deficiency diagnosis [26]. A recent published report has shown that Hb concentration is the key indicator for iron deficiency anemia [27]. In addition, RDW is also considered a highly effective and sensitive tool for early diagnosis of iron deficiency anemia [28]. Our data show that these various markers of anemia are significantly associated with LRG1, suggesting that LRG1 might be considered a potential iron deficiency marker. We also report a trend of increase in plasma LRG1 levels in anemic participants that is borderline significant (*p* = 0.07) compared to non-anemic adolescents. However, this needs to be confirmed in studies from other population groups in a larger cohort. LRG1 has also been negatively correlated with Hb levels in kidney transplant patients, possibly due to a secondary anemia induced by decreased renal function [1]. In this study, we observed significant associations between several iron deficiency markers and LRG1. However, due to the cross-sectional nature of this study, it cannot be deduced whether increased LRG1 is involved in the pathogenesis of anemia or is the consequence of anemia. While LRG1 has been previously reported to be involved in several health conditions [9], we herein describe, for the first time, its involvement in iron deficiency anemia.

It has been stated that iron deficiency increases HIF-1α protein expression [15,29], which upregulates the expression of IL-6 and TNFα [30,31,32]. Several pro-inflammatory cytokines, including IL-6 and TNFα, have been shown to regulate the expression of LRG1 at the transcriptional level by directly targeting LRG1 promoter [9]. This indicates that the LRG1 upregulation observed in children diagnosed with anemia might be a consequence of iron deficiency anemia. Hence, the proposed mechanism suggests that iron deficiency anemia might indirectly regulate LRG1 expression, possibly by upregulating HIF-1α expression, which regulates direct targets of LRG1 promoter including IL-6 and TNFα. Similarly, chronically stable hemodialysis patients also had elevated LRG1 levels, which were positively associated with inflammatory markers including IL-6 and hsCRP [33]. While this study did not mention if the patients had secondary anemia, they did find that patients with the highest LRG1 tertile also had significantly lower levels of hemoglobin and albumin [33], similar to the association observed in our study. This also adds to the debate of whether LRG1 is the cause or the consequence of anemia, since the chronic inflammation observed in the study is closely linked to CKD (chronic kidney disease) [34]. Nonetheless, it cannot be ruled out that the decreased anemia markers (hemoglobin and albumin) reported in this study [33] might also have contributed to the upregulation of the inflammatory markers, leading to an increase in LRG1 levels.

One of the strengths of our study is that it is one of the first studies to explore the association between LRG1 and iron deficiency as this association has not been extensively explored yet. In addition, our study investigated the association between LRG1 and iron deficiency in children, as establishing this association in a relatively healthy young population guarantees its independence from the complications commonly observed with adulthood iron deficiency. Moreover, our study was carried out on a reasonably large sample size and several statistical tools were employed to robustly show the association between LRG1 and iron deficiency in children. The main limitation of the study is the cross-sectional design, which prevented us from establishing causality. Therefore, we plan to perform a longitudinal study to further explore the association between LRG1 and iron deficiency anemia.

In conclusion, our findings demonstrate for the first time the strong association between iron deficiency anemia markers and LRG1 in otherwise healthy school-aged children. While LRG1 has been shown to be associated with several health conditions, its association with iron deficiency anemia, especially in children, has not been previously explored. Further studies are needed to corroborate those results and to look for similar associations in other population subgroups.

## Figures and Tables

**Figure 1 nutrients-15-03100-f001:**
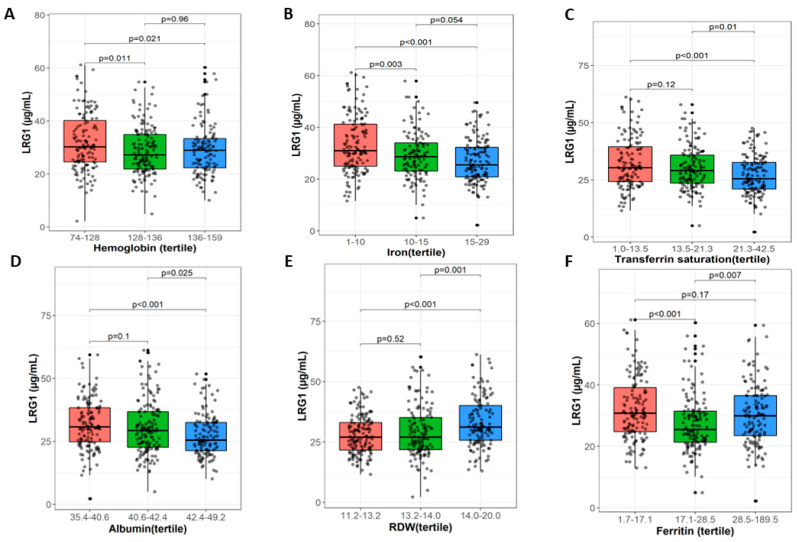
LRG1 plasma levels and iron-related markers.

**Figure 2 nutrients-15-03100-f002:**
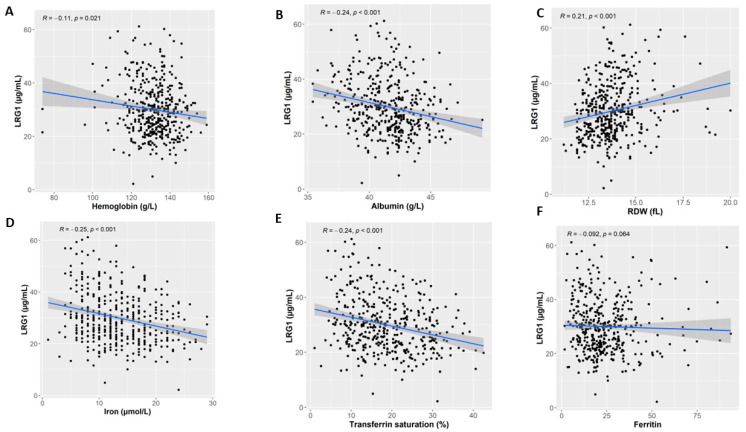
Correlation between LRG1 plasma level and iron anemia markers. (**A**) Hemoglobin, (**B**) albumin, (**C**) RDW, (**D**) iron, (**E**) transferrin, and (**F**) ferritin.

**Table 1 nutrients-15-03100-t001:** Characteristics of 431 adolescents enrolled in the study.

Characteristic	N = 431 ^1^
Sex	
Female	234 (54%)
Male	197 (46%)
WBC (×10^9^ cells/L)	6.70 (5.50, 8.19)
Hb (g/dL)	132 (126, 139)
Hct (L/L)	0.404 (0.390, 0.419)
RBC (×1012 cells/L)	5.02 (4.79, 5.26)
Gluc (mmol/L)	4.80 (4.40, 5.30)

^1^ n (%); Median (IQR).

**Table 2 nutrients-15-03100-t002:** The statistical comparison of LRG across tertiles of iron, transferrin saturation, albumin, RDW, and WBC.

Characteristic	LRG, N = 408 ^1^	*p*-Value ^2^
Iron (µg/dL)		<0.001
1–10	31 (25, 41)	
10–15	29 (23, 34)	
15–29	25 (21, 32)	
Transferrin saturation (in %)		<0.001
1.0–13.5	30 (24, 40)	
13.5–21.3	29 (24, 36)	
21.3–42.5	25 (21, 33)	
Albumin g/L		<0.001
35.4–40.6	31 (25, 38)	
40.6–42.4	29 (23, 37)	
42.4–49.2	25 (21, 32)	
RDW (fL)		<0.001
11.2–13.2	27 (22, 33)	
13.2–14.0	27 (22, 35)	
14.0–20.0	31 (26, 40)	
Ferritin (µg/L)		<0.001
1.7–17.1	31 (25, 39)	
17.1–28.5	25 (21, 31)	
28.5–189.5	30 (24, 37)	
WBC (10^9^/L)		0.004
3.1–5.9	27 (22, 34)	
5.9–7.7	29 (23, 35)	
7.7–12.8	31 (25, 38)	

^1^ Median (IQR); ^2^ Kruskal–Wallis rank sum test.

## Data Availability

The datasets used and/or analyzed during the current study are available from the corresponding author on reasonable request.

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
