# Peer review of "LRG1 Associates with Iron Deficiency Anemia Markers in Adolescents"

_nutrients, 2023, doi:10.3390/nu15143100_

Round 1
Reviewer 1 Report
The brief report “LRG1 Associates with Iron Deficiency Anemia Markers in Adolescents” aims at correlating LRG1 levels to iron deficiency in children (11-14 years).
Although the study is interesting, several points must be improved.
1- The group of participants enrolled in the study are not numerous and it is not specified whether females already with menses were excluded or considered separately. Is there any difference between Hb or other hematological parameters between males and females? If not, it should be clarified by showing the mean of each parameter.
2- Moreover, the author should include how many children show anemia among the 431 participants.
3- In addition to SI, %Transferrin Saturation or RDW did the authors analyze erythroid parameters (MCV, MCH…)? How did the authors establish the presence of iron deficient anemia? How did they exclude inflammatory related condition or other erythroid defects? Did the authors evaluate Epcidin levels? Did the authors evaluate some inflammatory markers (e.g IL-6)?
Author Response
June 11, 2023
Nutrients Journal
Manuscript ID: nutrients-2429839
Title: LRG1 Associates with Iron Deficiency Anemia Markers in Adolescents
To: Editor of Nutrients Journal,
First, we would like to sincerely thank you and the reviewers for your valuable time and useful contributions. We appreciate your input which helped improve our manuscript. Below you will find our point-by-point response to the referees and the changes we made based on the reviewers’ comments.
Reviewer 1
Comments and Suggestions for Authors
The brief report “LRG1 Associates with Iron Deficiency Anemia Markers in Adolescents” aims at correlating LRG1 levels to iron deficiency in children (11-14 years).
Although the study is interesting, several points must be improved.
- The group of participants enrolled in the study are not numerous and it is not specified whether females already with menses were excluded or considered separately.
We thank the reviewer for the positive comments. In the whole cohort (total sample 1451; 721 females; 59% of girls had reached menarche. Girls with menarche had significantly lower ferritin level compared to girls w/o menarche. However, when adjusted for ferritin, the difference in prevalence of anemia between boys and girls did not change (Shaban et al. Scientific Reports | (2020) 10:5857). Girls with menses were thus not excluded.
Is there any difference between Hb or other hematological parameters between males and females? If not, it should be clarified by showing the mean of each parameter.
As advised by the reviewer, we checked for gender-specific differences in the levels of RBC, WBC, Hb, Hct and Gluc. Of these five markers, only Hb and RBC showed sex-specific differences at a P-value < 0.05.
Supplementary Table 1. Hematological parameters differences based on gender.
|
Characteristic |
Female, N = 2341 |
Male, N = 1971 |
p-value2 |
|
RBC (×1012cells/L) |
4.99 (4.73, 5.21) |
5.10 (4.86, 5.37) |
0.003 |
|
WBC (×109 cells/L) |
6.70 (5.70, 8.30) |
6.50 (5.34, 8.10) |
0.14 |
|
Hb (g/dl) |
131 (125, 136) |
134 (127, 140) |
<0.001 |
|
Hct (L/L) |
0.403 (0.389, 0.416) |
0.408 (0.390, 0.422) |
0.089 |
|
Gluc (mmol/L) |
4.80 (4.50, 5.30) |
4.90 (4.40, 5.40) |
0.8 |
|
1Median (IQR) |
|||
|
2Wilcoxon rank sum test |
|||
The following sentences were added to the result section in lines 8-11 of page 3:
“We also checked for gender-specific differences in the levels of RBC, WBC, Hb, Hct and Gluc. Of these five markers, only Hb and RBC showed sex-specific differences at a P-value < 0.05 (Supplementary Table 1).”
- Moreover, the author should include how many children show anemia among the 431 participants.
The requested data is presented below.
Supplementary Table 2. The distribution of anemia subjects based on gender.
|
Characteristic |
Female, N = 2341 |
Male, N = 1971 |
p-value2 |
|
|
|
|
0.3 |
|
Anemic |
21 (9.0%) |
12 (6.1%) |
|
|
Not-anemic |
212 (91%) |
185 (94%) |
|
|
1n (%) |
|||
|
2Pearson's Chi-squared test |
|||
The following comment was added to the result section in lines 11-13 of page 3:
“We have also assessed the distribution of anemia subjects based on gender (Supplementary Table 2) and the result was not significant.”
- In addition to SI, %Transferrin Saturation or RDW did the authors analyze erythroid parameters (MCV, MCH…)? How did the authors establish the presence of iron deficient anemia? How did they exclude inflammatory related condition or other erythroid defects? Did the authors evaluate Epcidin levels? Did the authors evaluate some inflammatory markers (e.g IL-6)?
Based on previous paper on anemia in this cohort (Shaban et al. Scientific Reports | (2020) 10:5857), Fe deficiency anemia was suggested for three reasons: (1) Fe levels were lower in anemic vs non-anemic, (2) MCV were lower in anemic compared to non-anemic, thus suggesting microcytic RBC, (3) Low ferritin and transferrin saturation in anemic vs, non-anemic. Furthermore, we did not find any association of B12 and folate levels with anemia, and none of the children had any genetic disease related to hemoglobinopathies.
Since we do not have the values for MCH, we performed our analysis only on MCV values. Results on examining associations of Hb, CRP, and MCV with LRG are presented in the following table. MCV did not associate significantly with LRG.
The following sentence was added to the result section in lines 31-33 of page 3:
“We also assessed the associations between MCV and LRG and the result was not statis-tically significant (Supplementary Table 3).”
Supplementary Table 3. Statistical comparison of LRG across tertiles of CRP, Hb and MCV.
|
Characteristic |
LRG |
p-value2 |
|
Hb |
|
0.019 |
|
74.0-128.0 |
30 (25, 40) |
|
|
128.0-136.0 |
27 (22, 35) |
|
|
136.0-159.0 |
29 (22, 33) |
|
|
CRP |
|
<0.001 |
|
0-0.28 |
27 (21, 33) |
|
|
0.28-1.54 |
27 (22, 34) |
|
|
1.54-9.89 |
30 (26, 37) |
|
|
MCV (fL) |
|
0.5 |
|
56.4-78.6 |
30 (22, 37) |
|
|
78.6-83.2 |
30 (24, 36) |
|
|
83.2-106.5 |
28 (22, 36) |
|
|
1Median (IQR) |
||
|
2Kruskal-Wallis rank sum test |
||
Furthermore, none of the children in the whole cohort had any known inflammatory conditions or diagnosed hemoglobinopathies. These were all apparently healthy adolescents.
As for the hepcidin levels, it was not done in our cohort and the participants were not phenotyped for it. Moreover, we attempt to evaluate IL-6 levels in the cohort. Since the sensitivity of the the assay was very low and more than 60% of the participants were having IL-6 levels at 0.03 ng/ml. So, it was hard to perform any analysis on IL-6. Analysis on CRP value is now given in the tables above.
Reviewer 2 Report
The study LRG1 Associates with Iron Deficiency Anemia Markers in Adolescents is an short report for determine the association between iron deficiency anemia and LRG1.
Some comments for the authors:
1. Line 53 – HIF is not explained as an abbreviation
2. It is not clear the number of children with anemia and without it. A comparation between the characteristics of these two groups should be presented and if the groups are similar.
3. I do not seem to find the figures. Could you insert it in the main manuscript?
4. The three groups (tertiles) of Hb are not well defined into the main article.
5. What is the practical use of your results?
6. In table 2 it is not clear the number 408. What is stands for?
7. You suggest that LRG1 should be used for diagnosis of anemia? Why do we need another expensive marker for anemia diagnosis?
8. It was suggested that HIF is involved in ferroportin regulation. Are the some indication that LRG1 is involved in iron homeostasis?
9. Is is possible the the children had other condition that could influence your results?
Author Response
June 11, 2023
Nutrients Journal
Manuscript ID: nutrients-2429839
Title: LRG1 Associates with Iron Deficiency Anemia Markers in Adolescents
To: Editor of Nutrients Journal,
First, we would like to sincerely thank you and the reviewers for your valuable time and useful contributions. We appreciate your input which helped improve our manuscript. Below you will find our point-by-point response to the referees and the changes we made based on the reviewers’ comments.
Reviewer 2
Comments and Suggestions for Authors
The study LRG1 Associates with Iron Deficiency Anemia Markers in Adolescents is an short report for determine the association between iron deficiency anemia and LRG1.
Some comments for the authors:
- Line 53 – HIF is not explained as an abbreviation.
HIF was detailed in line 8 of page 2.
- It is not clear the number of children with anemia and without it. A comparation between the characteristics of these two groups should be presented and if the groups are similar.
21 of 234 female children and 12 of 197 male children were anemic and the rest non-anemic.
Comparison of the various clinical characteristics between the anemic and no- anemic children are given in the below table:
Supplementary Table 4. Comparison of various clinical characteristics based on anemia status.
|
Characteristic |
Anemic, N = 331 |
Not-anemic, N = 3971 |
p-value2 |
|
Gluc (mmol/L) |
5.00 (4.70, 5.40) |
4.80 (4.40, 5.30) |
0.14 |
|
Albumin g/liter |
40.10 (38.50, 41.80) |
41.60 (40.00, 42.83) |
0.002 |
|
Iron (µg/dL) |
9.0 (6.0, 13.0) |
13.0 (9.0, 17.0) |
0.002 |
|
Transferrin saturation (in %) |
13 (7, 20) |
17 (13, 24) |
0.010 |
|
Ferritin (µg/L) |
13 (6, 33) |
21 (15, 31) |
0.021 |
|
WBC (×109 cells/L) |
6.70 (5.70, 7.70) |
6.68 (5.50, 8.20) |
0.8 |
|
RBC (×1012cells/L) |
4.88 (4.52, 5.52) |
5.02 (4.81, 5.25) |
0.6 |
|
Hb (g/dl) |
113 (107, 117) |
133 (128, 139) |
<0.001 |
|
Hct (L/L) |
0.363 (0.347, 0.370) |
0.407 (0.394, 0.420) |
<0.001 |
|
MCV |
71 (65, 79) |
81 (78, 85) |
<0.001 |
|
RDW (fL) |
15.70 (13.90, 17.20) |
13.50 (12.90, 14.20) |
<0.001 |
|
CRP |
0.29 (0.12, 2.54) |
0.65 (0.17, 2.10) |
0.5 |
|
LRG ug/mL |
31 (25, 42) |
29 (23, 35) |
0.07 |
|
1Median (IQR); n (%) |
|||
|
2Wilcoxon rank sum test; Pearson's Chi-squared test; Fisher's exact test |
|||
|
|
|||
- I do not seem to find the figures. Could you insert it in the main manuscript?
As suggested by the reviewer, we have added the figures to the end of the main manuscript.
- The three groups (tertiles) of Hb are not well defined into the main article.
Based on the reviewer advise, we have defined Hb group tertiles in supplementary Table 2.
- What is the practical use of your results?
We would like to address your comment regarding the practical relevance of our study results. We think that enhancing understanding is a crucial practical implication of our findings. By investigating the association between leucine-rich α-2 glycoprotein 1 (LRG1) and iron deficiency anemia markers in children, our study contributes to a deeper understanding of the underlying mechanisms involved in this condition. Moreover, it is worth noting that our study was conducted on a substantial sample size, which enhances the robustness of our results. We employed various statistical tools to rigorously demonstrate the association between LRG1 and iron deficiency in children. Our findings contribute to the field by deepening the understanding of the association between LRG1 and iron deficiency anemia, which can improve diagnostic criteria, enable targeted interventions and personalized medicine, and inform prevention and public health strategies.
Thank you for your valuable comments and suggestions.
- In table 2 it is not clear the number 408. What is stands for?
The analysis was conducted on 431 participants. This number will be corrected in table 2.
- You suggest that LRG1 should be used for diagnosis of anemia? Why do we need another expensive marker for anemia diagnosis?
We appreciate your comment regarding the potential use of LRG1 as a diagnostic marker for anemia. While our study suggests a strong association between LRG1 and iron deficiency anemia markers in children, we acknowledge the practical considerations surrounding the implementation of a new marker for diagnosis. It is important to note that our study does not propose LRG1 as a replacement for existing diagnostic markers of anemia. Rather, it highlights the potential of LRG1 as an additional marker that may provide valuable information in the context of anemia diagnosis, particularly in children. We agree that the adoption of a new marker for anemia diagnosis should be carefully considered in terms of its cost-effectiveness and potential benefits. While LRG1 shows promise as an additional marker, further research is needed to evaluate its clinical utility and compare it with existing diagnostic markers to determine its potential role in enhancing the accuracy of anemia diagnosis.
- It was suggested that HIF is involved in ferroportin regulation. Are the some indication that LRG1 is involved in iron homeostasis?
We appreciate the reviewer comment. However, we cannot make any inference from this study without looking at hepcidin levels.
- Is it possible the the children had other condition that could influence your results?
We appreciate your question regarding the possibility of other conditions influencing our study results. While it is true that the presence of confounding factors is always a potential concern in any study, we took several steps to minimize their impact on our analysis. Firstly, none of the children in our study had conditions related to hemoglobinopathy, such as thalassemia or sickle-cell anemia. By excluding these specific conditions, we aimed to reduce the influence of confounding factors that could directly impact hemoglobin levels and potentially introduce bias into our analysis.

Round 2
Reviewer 1 Report
thank you for the reply
Reviewer 2 Report
The article can be publish in the present form.